# Eyes Show the Way: Modelling Gaze Behaviour for Hallucination Detection

**Kishan Maharaj**[* †], **Ashita Saxena**[* †], **Raja Kumar**[†],
**Abhijit Mishra**[‡], **Pushpak Bhattacharyya**[†]

[†]Indian Institute of Technology Bombay, Mumbai, India
[‡]University of Texas at Austin, Texas, United States

{kishan.maharaj.iitb, as.saxena.as, kumar.raja.iitb}@gmail.com[†],
pb@cse.iitb.ac.in[†], abhijitmishra@utexas.edu[‡]

## Abstract

Detecting hallucinations in natural language processing (NLP) is a critical undertaking that demands a deep understanding of both the semantic and pragmatic aspects of languages. Cognitive approaches that leverage users' behavioural signals, such as gaze, have demonstrated effectiveness in addressing NLP tasks with similar linguistic complexities. However, their potential in the context of hallucination detection remains largely unexplored. In this paper, we propose a novel cognitive approach for hallucination detection that leverages gaze signals from humans. We first collect and introduce an eye tracking corpus *(IITB-HGC: IITB-Hallucination Gaze corpus)* consisting of 500 instances, annotated by five annotators for hallucination detection. Our analysis reveals that humans selectively attend to relevant parts of the text based on distributional similarity, similar to the *attention bias* phenomenon in psychology. We identify two attention strategies employed by humans: *global attention*, which focuses on the most informative sentence, and *local attention*, which focuses on important words within a sentence. Leveraging these insights, we propose a novel cognitive framework for hallucination detection that incorporates these attention biases. Experimental evaluations on the FactCC dataset demonstrate the efficacy of our approach, obtaining a balanced accuracy of 87.1%. Our study highlights the potential of gaze-based approaches in addressing the task of hallucination detection and sheds light on the cognitive processes employed by humans in identifying inconsistencies.

## 1 Introduction

Hallucination detection in text refers to the task of identifying and validating information that is inaccurately or falsely represented within textual content. Detection of hallucination involves examining the claims made in the text and assessing their alignment with the surrounding context and external knowledge. Addressing hallucinations has become paramount, particularly in the context of automatically generated text utilizing powerful language models (LLMs), which often exhibit human-like fluency but are prone to hallucinatory outputs (Zhang et al., 2023; Alkaissi and McFarlane, 2023).

Many existing methods for hallucination detection depend on knowledge sources that are explicit such as Wikipedia or knowledge graphs (Manakul et al., 2023; Santhanam et al., 2021; Dziri et al., 2021; Ji et al., 2023) or ingrained in language encoders such as BERT or RoBERTa (Shen et al., 2023; Zhou et al., 2020). While these traditional approaches can reasonably detect hallucinations in a text when supplemented with knowledge sources, they face sustainability challenges due to the constant need for up-to-date knowledge. Obtaining the latest information for hallucination detectors is often impractical, as it requires readily available and current sources of knowledge. To address this issue, we propose an alternative approach that leverages cognitive and behavioural information from humans in the form of gaze patterns while they analyze text for potential hallucinations.

Our work is motivated by the notion that humans, while reading text for hallucination identification would naturally employ their cognitive faculties to navigate the intricate relationship between language and real-world knowledge. Linguistically, this involves scrutinizing whether (a) entities in the text are adequately placed (e.g., is *Canada* a right choice of entity) (b) the semantic roles played by the entities are valid w.r.t the context (e.g, *Pluto is a planet*). This critical examination would often manifest as prolonged fixations[1] on specific sections of text (e.g., longer fixations on entities and phrases such as Canada, Pluto and a planet) that require closer evaluation, resulting in denser and

---

[*]These authors contributed equally to this work

[1]A fixation occurs when the eye is focused on a particular part of the screen

more extensive fixation activities. In essence, fixations may serve as invaluable indicators, acting as a reliable surrogate for knowledge-based validation of contextual information pertaining to potential hallucinations. This may open up possibilities for the development of a hallucination detector that can leverage gaze data as a primary input, alleviating the reliance on supplementary external knowledge.

To validate this, we first collect a first-of-its-kind eye-tracking data of 5 annotators annotating 500 instances of claim-context pairs, carefully derived from the FactCC dataset (Kryscinski et al., 2020). During the annotation process, we capture the fixation patterns of annotators on both the claim and context texts, along with their corresponding labels. Notably, our inter-annotator agreement (IAA) Kappa score reaches 0.60, indicating substantial agreement. Behavioural analysis of the annotated data reveals a recurrent pattern where annotators tend to skim through somewhat irrelevant context while selectively focusing on information crucial for establishing or refuting hallucinations. Building upon insights gained from this behavioural analysis, we term this selective reading phenomenon as *"attention bias"*. Furthermore, our observations indicate that attention bias can manifest as either a *"global"* approach, involving the extraction of sentences containing relevant information about hallucinations, or a *"local"* approach, focusing on specific phrases within sentences to evaluate the alignment of semantic roles between the claim and the selected phrases.

Building upon these insights, we propose a modular architecture that incorporates global and local attention bias using transformer-based deep learning techniques (Vaswani et al., 2017) and a gaze-based attention saliency module (Sood et al., 2020b). Experimental evaluations on the FactCC dataset demonstrate the efficacy of our approach, outperforming baseline models while attaining better interpretability. We have open-sourced our code, data and results for academic usage [2].

Our contributions are summarised as follows:

- We create and share a first-of-its-kind eye-tracking corpus (Section 3) for hallucination detection: IITB-HGC (IITB-Hallucination Gaze Corpus) consisting of 500 instances of context and claim pairs, where five annotators label the claim as hallucinated or non-hallucinated with respect to the context. We

---

[2]Github Link

obtained an IAA Kappa score of 0.60, which indicates substantial agreement.

- We introduce a novel concept of *attention bias* derived by analyzing human annotators' gaze patterns while they carry out the task of hallucination detection.

- We propose and evaluate a cognitively inspired BERT-based deep learning framework (Figure 2) for hallucination detection driven by different forms of attention biases seen in human reading. The framework offers superior performance (Table 2), and better interpretability (Figure 4) against baseline approaches and shows competitive performance with SOTA by attaining the balanced accuracy of 87.1% (Table 2).

## 2 Related Work

### 2.1 Hallucination / Inconsistency Detection

Prior work in understanding hallucination includes a survey by Ji et al. (2022) which discusses hallucination arising from data, training approach, and inference. Hallucination detection in various NLG tasks has been tackled previously using statistical methods (Wang et al. (2020b), Shuster et al. (2021)) and model-based methods. Model based methods include QA-based approaches (Scialom et al. (2021), Wang et al. (2020a), Honovich et al. (2021)) and NLI-based approaches (Kryscinski et al. (2020), Mishra et al. (2021), Laban et al. (2022a)). Recently, prompt based methods (Arora et al. (2022), Manakul et al. (2023), Agrawal et al. (2023), Dhuliawala et al. (2023)) are being used to detect hallucinations in the text produced by LLMs. Usage of gaze signals in hallucination detection remains an unexplored field, although there is prior work which uses gaze in other NLP tasks (as mentioned in Section 2.2).

Kryscinski et al. (2020) proposed a weakly-supervised model for verifying factual consistency and released a dataset FactCC which contains claim-context pairs. We use this dataset to conduct our experiments. *To the best of our knowledge, the usage of gaze signals has not been explored in the past for the task of hallucination detection.*

### 2.2 Gaze in deep learning-based architectures

In recent years, various attempts have been made to investigate the correlation of human attention with the machine attention of a pre-trained large

language model (Eberle et al. (2022), Sood et al. (2020a), Bensemann et al. (2022)). Eberle et al. (2022) highlighted the inability of cognitive models to account for the higher level cognitive activities like semantic role matching, hence motivating the use of large language models (LLMs) for modelling the human gaze. Hollenstein et al. (2021) showed the efficacy of LLMs in predicting the gaze features for multiple languages, including English, Russian, Dutch and German. Barrett et al. (2018) used natural reading eye-tracking corpus for regularizing attention function in a multi-task setting.

The work most similar to us is Sood et al. (2020b), which investigates the integration of the gaze-based text saliency model with vanilla transformers (Vaswani et al., 2017) for directly incorporating gaze predictions into the attention mechanisms for paraphrase detection and sentence compression tasks. However, their proposed architecture heavily relies on BiLSTM and vanilla transformers, lacking pre-training benefits. In contrast, our approach utilizes pre-trained language models and does not impose constraints on the attention functions of these language models. Instead, we focus on learning the space for contextual embedding transformation based on saliency modelling grounded by human gaze data.

## 3 Collection of IITB-HGC (IITB-Hallucination Gaze Corpus)

We collect eye-tracking data from annotators while they perform the task of hallucination detection. For the data collection process, we use 500 claim-context pairs from the FactCC dataset (Kryscinski et al., 2020) as stimuli. The annotators are given a claim-context pair and are asked to assess the faithfulness of the claim in relation to the context. **To the best of our knowledge, no eye-tracking corpus is available for hallucination detection.**

300 pairs are chosen from the training set, 100 from the validation set, and 100 from the test set of the FactCC dataset and are used in similar splits during the training of the local attention bias model to avoid data leaks. The acquisition of the IITB-Hallucination Gaze Corpus is then carried out in *three phases*, in which we presented annotators with 180, 160, and 160 instances respectively.

The average number of words in the claim-context pair is 105, the shortest claim having 6 words and the most extended claim having 32. On average, each claim consists of 13 words. The con-

text length ranges from a minimum of 56 words to 132 words, with 92 words per context on average.

### 3.1 Experimental Settings

Participants were provided with detailed task instructions, including two example claim-context pairs and their expected annotations, displayed on the computer screen for guidance. The annotation guidelines are described in Appendix B.2. Participants completed the tasks individually in a monitored room, assisted by a research assistant. More details about the experiments and other environmental settings for the eye-tracking data collection can be found in Appendix B.1.

### 3.2 Participant/Annotator details

The annotation process involved five participants aged 21-25 (two male, three female) with English as a primary language of academic instruction. Among them, one participant had a Bachelor's degree, one held a Doctorate, and three had Master's degrees from universities with English as the primary language of instruction. All the participants had valid TOEFL scores with a minimum of 100 and an average of 107.2 out of 120, demonstrating acceptable English proficiency.

**Figure 1:** Heatmap of human gaze fixation for a claim-context instance while performing the task of hallucination detection (red signifies higher values). Fixations are seen only on the most similar sentence. Relatively higher fixation is seen on the words responsible for hallucination (i.e., *"remains are discovered"*).

### 3.3 Behavioural Analysis

Participants exhibit a hierarchical approach when searching for and matching the claim within the context. They begin by thoroughly reading and encoding the claim into memory, either in its entirety or by noting key elements like numbers and proper nouns. Subsequently, they scan the context sentences from the first to the last, searching

for a sentence corresponding to their remembered claim or key elements. When a potential match is found, they focus on that specific sentence, engaging in a more detailed reading to verify its coherence with the claim. Notably, there is a strong bias towards sentences distributionally similar to the claim within the context. This bias can be linked to the initial lexical processing stage of the reading theory (Just and Carpenter, 1980). It is noteworthy to mention that semantic similarity's influence is less pronounced than distributional similarity. Given the nature of the task, where there is a possibility of semantic contradiction between the claim and the relevant sentence in the context, human annotators learn to rely on distributional similarity as a guiding factor. This behaviour of humans can be seen via heat maps of the fixation of the human gaze on context words. One such example is shown in Figure 1. It can be seen that the relevant sentence is fixated on the most number of times whereas the other sentences have zero/very-low fixation.

## 4 Methodology

Building upon our theory of attention bias, our proposed cognitive framework addresses the hallucination task by integrating this behaviour. Figure 2 provides an overview of the framework, featuring the modelling of global and local attention bias.

### 4.1 Attention Bias

Attention bias is a goal-driven cognitive mechanism in which humans while making decisions selectively focus on relevant information while disregarding irrelevant or less salient stimuli (Fadardi et al., 2016). They search for *distributionally* similar sentences and then verify the semantics at the token level. Figure 1 illustrates this behaviour. Upon deeper examination of gaze behaviour in IITB-HGC, two distinct types of attention bias were observed - *global* and *local* attention bias, which we describe in the following subsections.

#### 4.1.1 Modeling Global Attention Bias

Global attention bias represents the human inclination to prioritize the most informative sentence in a given context. By simulating human attention bias, our approach effectively identifies the most relevant context sentence for a given claim. To emulate this behaviour, we propose an ensemble approach that utilizes multiple cutting-edge

sentence-transformer models[3]. Specifically, we select the following models for computing similarity scores: *all-roberta-large-v1* (Liu et al., 2019), *all-mpnet-base-v1* (Song et al., 2020), *gtr-t5-large* (Ni et al., 2021), *all-mpnet-base-v2* (Song et al., 2020), and *LaBSE* (Feng et al., 2020). These models are chosen for their ability to generate high-quality sentence embeddings while maintaining moderate model sizes. Appendix (Section A.1) gives details on their performances and model sizes.

To identify the optimal mechanism for utilizing similarity scores from various models, we employ the following methodology. Using IITB-HGC (detailed in Section 3), we leverage fixation data collected during the claim labelling process by humans to extract the sentence with the longest fixation duration. This sentence, serving as the ground truth for our global attention model, guides our objective of determining the optimal combination of similarity scores from five different models based on the most fixated sentence by humans. Upon analyzing the results, we find that a voting mechanism yields the highest accuracy for these 500 instances (Table 5). This voting mechanism involves selecting the sentence that receives the highest similarity scores from the majority of the models. Based on a manual analysis, we find that the voting mechanism consistently captures the correct relevant sentence, surpassing other alternatives. As a result, we adopt the voting-based ensembling of sentence encoders as our global attention bias model. Interestingly, we also observe that using the sentence with the maximum score from only the LaBSE model achieves the second-best accuracy. Thus, LaBSE model can also be used as an alternative cost effective approach. In summary, given a claim and context, our pipeline first employs the ensemble global attention model to identify the most relevant sentence. This sentence is then utilized as input for the subsequent steps in our end-to-end hallucination detection model (Figure 2). Notably, the global attention bias model also helps in reducing the number of tokens given as input to the subsequent model. Figure 6 in Appendix Section A.2 shows a visual representation of how the global attention bias model is implemented.

#### 4.1.2 Modeling Local Attention Bias

Local attention bias refers to the annotator's inclination to prioritize salient words essential for

---

[3]Documentation for sentence-transformer models

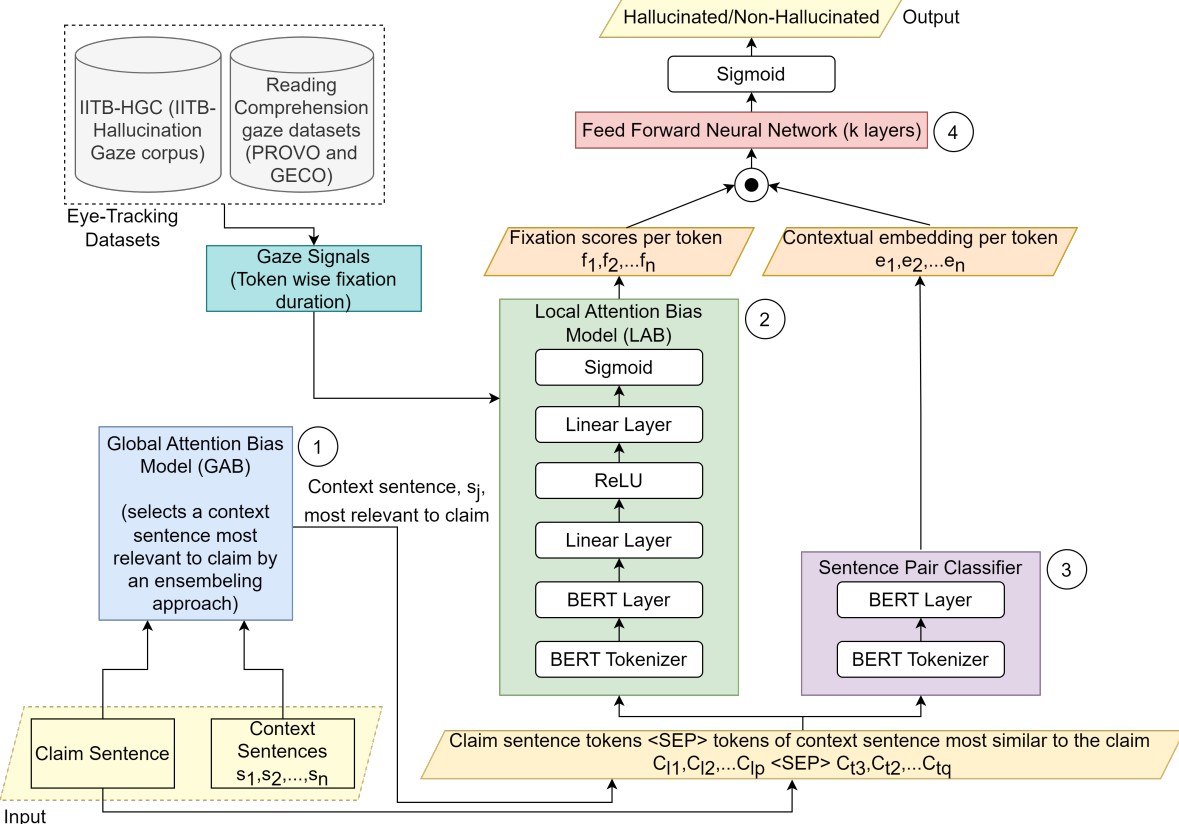

**Figure 2:** Overview of the end-to-end model incorporating global attention and local attention bias for detecting hallucination given a claim and a context. Figure 6 shows the details of the global attention bias model (1).

hallucination detection, while disregarding irrelevant words. Modeling local attention bias provides a means to influence the self-attention components (Vaswani et al., 2017) of transformer-based deep encoders (Devlin et al., 2018) employed in downstream tasks such as hallucination classification. This guided attention approach enhances token-level model interpretability (see Section 6).

We model the local attention bias by learning the token level saliency score using eye-tracking data in a separate module called Local Attention Bias. The LAB module is jointly trained alongside the sentence pair classifier to adapt to the task-specific parameters. The overall flow of information is depicted in Figure 2 in blocks labelled (2), (3) and (4) and the training process is described below.

Let the $C_l$ and $C_t$ be the vectors representing the claim and context pair respectively:

$$C_l : \{l_1, l_2, l_3, ....l_p\}, C_t : \{t_1, t_2, t_3, ....t_q\} \quad (1)$$

The input vector is the concatenation of claim and context separated by a special [SEP] token.

$$I : [C_l; [SEP]; C_t] \quad (2)$$

The output vector of the LAB model, representing token level importance scores, is defined as:

$$F_{lab} : \{f_1, f_2, f_3, ....f_{p+q+1}\} \quad (3)$$

The first phase of the training involves learning the normalized fixation duration at the token level from the gaze data. These learned parameters are checkpointed and used to instantiate the local attention bias model in the next phase of training. The next phase of training involves the joint learning of the hallucination detection model (sentence pair classifier) and local attention bias model. This phase begins with the initialization of the local attention bias model with the parameters learned in the previous phase and proceeds with the training by hallucination detection data on a single objective function optimized with binary cross entropy loss on the final output. The output of the final model can be defined mathematically as :

$$Y = Sigmoid(W \cdot (F_{lab} \circ E_{embed}) + b) \quad (4)$$

Here $W$ and $b$ are the parameters of the feed-forward network, $E_{embed}$ represents the contextual embedding and $Y$ represents the final output. By

incorporating both global and local attention biases, we aim to simulate the cognitive processes observed in humans during hallucination detection.

## 5 Experiments

### 5.1 Dataset Details

**FactCC Dataset:** We use the FactCC dataset released by Kryscinski et al. (2020) which consists of ~1M instances. The data distribution is mentioned in Table 1. Kryscinski et al. (2020) prepare this dataset by introducing perturbations (e.g., entity swapping, pronoun swapping, negation, back translation etc.) in a sentence taken from the context. We specifically selected the FactCC dataset for our study for several compelling reasons. Firstly, the dataset exhibits a wide range of topics, providing diverse contexts to analyze. This diversity ensures that our experimental design captures a broad spectrum of claim-context relationships. Secondly, the FactCC dataset includes well-defined perturbations, which can be leveraged to precisely examine the process of verifying claims based on the given context which facilitated a clear and controlled experimental setup for our eye-tracking experiment.

| Split | H | NH | Total |
|---|---|---|---|
| Train | 499,623 | 503,732 | 1,003,355 |
| Test | 62 | 441 | 503 |
| Validation | 132 | 799 | 931 |
| Total | 499,817 | 504,972 | 1,004,789 |

**Table 1:** Distribution of different types of instances (hallucinated (H) and non-hallucinated (NH)) in different splits of FactCC dataset (train, test and validation).

**Eye-Tracking Corpus:** This is the curated eye-tracking corpus (IITB-Hallucination Gaze Corpus) we explained in Section 3. The corpus is divided into the train, validation, and test sets, with a 60%, 20%, and 20% split, respectively and used as the source of supervision for the local and global attention models. It should be noted that the purpose of gaze data was to do a behavioural analysis of humans during hallucination detection. This makes 500 instances enough to gather and validate the insights from this study, which were used to design the proposed cognitive framework and ground the global attention bias model. However, for finetuning the local attention bias model, we leverage two other published datasets for pre-training our local

attention models - the **PROVO** Corpus [4](Luke and Christianson, 2018) and the **GECO** corpus[5] (Cop et al., 2017). These datasets are collected for the task of reading comprehension and encompass gaze patterns during general-purpose reading, which we believe would be beneficial in effectively initializing our local attention model. In Appendix E, we show a word-level comparison with other publicly available gaze datasets.

### 5.2 Experimental Details

In our experimental setup, we conducted training using a single NVIDIA A100-SXM4-80GB GPU with batch size of 128, and a learning rate of 2e-5 was used during the training process. To capture the majority of the FactCC data, we set the sequence length to 320, covering approximately 95% of the dataset. For training, we used the Binary Cross Entropy Loss as the loss function and utilized the Adam optimizer with a weight decay of 1e-2.

At run-time, the detection of hallucination happens as follows. The most relevant context sentence is first selected using global attention bias, aiding claim verification. The output of the global attention bias model, combined with the claim, is passed to the local attention bias model trained on eye-tracking data. The local attention bias model assigns fixation scores to each token, indicating their saliency for hallucination detection. Simultaneously, the sentence pair classifier model generates contextual embeddings for each token. The embeddings are then weighted using the fixation scores and passed through the feed-forward and sigmoid layers to generate output class labels. Training of the model employs a similar forward flow of information as above, and the updation of weights happens through backpropagation, details of which are skipped for brevity.

## 6 Results

Table 2 displays our experimental results in different experimental settings as described below:

**BERT+GAB**: BERT architecture integrated with Global Attention Bias model.

**BERT+GAB+LAB**: BERT architecture integrated with Global Attention Bias and Local Attention Bias models.

**BERT+GAB+LAB+GAZE**: BERT architecture integrated with Global Attention Bias and Local

---

[4]https://osf.io/sjefs
[5]https://expsy.ugent.be/downloads/geco/

| Model | Balanced Accuracy |
|---|---|
| BERT MNLI | 0.5151 |
| BERT FEVER | 0.5207 |
| NER Overlap | 0.55 |
| MNLI-doc | 0.613 |
| FactCC-CLS | 0.759 |
| DAE | 0.759 |
| FEQA | 0.536 |
| QuestEval | 0.666 |
| ChatGPT-ZS | 0.747 |
| ChatGPT-COT | 0.795 |
| SummaCZS | 0.838 |
| **SummaC-Conv** | **0.895** |
| BERT+GAB* | 0.69455 |
| BERT+GAB+LAB* | 0.86371 |
| **BERT+GAB+LAB +Gaze***| **0.87103** |

**Table 2:** Comparison of our work (marked with *) with baselines. Here, GAB and LAB refer to Global Attention Bias and Local Attention Bias respectively. BERT+GAB+LAB+Gaze refers to the model with global and local attention bias trained on the complete gaze data (PROVO+GECO+IITB-HGC)

Attention Bias finetuned on complete gaze data.

We observe improvements by incorporating global attention bias over the baselines, which highlights the importance of noise filtering in foundational large language models. Furthermore, we observed significant enhancements over the previous setting after incorporating the local attention bias model. Notably, the local attention bias model demonstrated further improvements upon the introduction of gaze data. However, both the McNemar and t-tests report no statistically significant difference between the two model outputs as the p-values never go below 0.05. Table 2 compares our model with all the baselines described in the Appendix C. We observe that the proposed approach shows significant improvements over the baselines (including ChatGPT-ZS and ChatGPT-COT)[6] and competitive results with the state-of-the-art (Laban et al., 2022b).

## 7 Analysis

We perform analysis on the fixation data from IITB-HGC and present it in Section 7.1. We also show that including complete gaze data increases inter-

[6]https://openai.com/blog/chatgpt

pretability in Section 7.2. We perform an error analysis (Section D) to understand the scenarios in which our model gives incorrect responses. Our comparative study with the SummaC-Conv model (Section 7.4) shows that our model performs better in 92% of the cases in which SummaC-conv makes erroneous predictions.

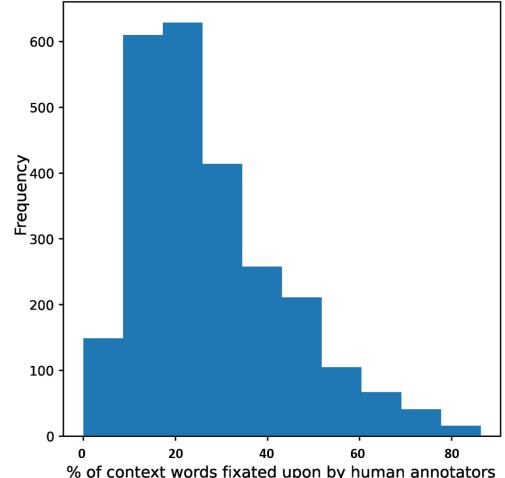

**Figure 3:** This figure shows a histogram depicting the percentage of context words that are being fixated upon by annotators over the IITB-HGC. It can be seen that only 10-30% of context words are being fixated the most number of times.

| Error Category | No. of Instances (%) | Instances where our model is correct (%) |
|---|---|---|
| Exact claim | 34 (45.9%) | 34/34 (100%) |
| Paraphrased claim | 19 (25.7%) | 18/19 (94.73%) |
| Ambiguous claim | 7 (8.1%) | 7/7 (100%) |
| Coreference in claim | 8 (9.5%) | 8/8 (100%) |
| Incorrect true label | 7 (10.8%) | 2/7 (37.5%) |

**Table 3:** The distribution of each type of SummaC-Conv error category. The last column shows the number of instances in which our model gives correct prediction for each error category.

### 7.1 Fixation Analysis

Figure 3 shows the percentage of words being fixated upon by annotators in the IITB-HGC. For each instance, we compute the percentage of words that are being fixated upon. It is observed that the majority of words in each instance are not fixated upon at all. Figure 3 shows that only 10-30% of the words

**(a)** Heatmap for BERT + global attention bias + local attention bias model finetuned on gaze data. The predicted label is 'Hallucinated'

**(b)** Heatmap for BERT + global attention bias + local attention bias model without gaze data fine-tuning. The predicted label is 'Hallucinated'

**(c)** Heatmap for BERT + global attention bias based on raw attentions. The predicted label is 'Not Hallucinated'

**Figure 4:** The figure displays heatmaps depicting the local attention bias scores for all three settings for an example marked as 'hallucinated'. Darker colours indicate higher scores, while lighter colours indicate lower scores.

in the entire IITB-HGC are fixated the most number of times. We also compute the sentence-wise fixation for every instance. We observe that on average, for every instance, 78% of the total fixation is on only 1-2 sentences of the context. Other sentences have very less or no fixation.

## 7.2 Interpretability Analysis

Let us consider the following example:

*Claim*: *"she was the one who was remembered at his memorial service by these words: " big heart, big smiles, big service"*

*Context*: *"and for Sean Collier, who was remembered at his memorial service by these words: " big heart, big smiles, big service"*

The above claim is hallucinated with respect to the context due to the entity replacement of "Sean Collier" with the pronoun "she". Figure 4 displays heatmaps depicting the importance of each words. We observe that the LAB model correctly highlights the pronoun "she" in both the scenarios of with gaze and without gaze. In comparison, after removing the LAB bias model the word "and" is incorrectly highlighted and the word "she" is not highlighted at all. Moreover, the signal strength on important words ("she" in this example) weakens after removing the gaze and completely goes away after removing the LAB module.

## 7.3 Error Analysis

Significant errors in predictions are observed due to the high semantic overlap between the claim and context which were labelled as hallucinated.

For example:

*Claim*: *"The movie is first in the rampaging-dino franchise since "Jurassic Park iii" Pratt's scientist Chris Pratt is the movie's new trailer."*

*Context*: *"The movie is first in the rampaging-dino franchise since "Jurassic Park III" in 2001."*

The true label of the current example is "Hallucinated" while the prediction by our model is "Not Hallucinated". We can see that there is a significant semantic overlap between the claim and context which could have resulted in the prediction of "Not Hallucinated". It is interesting to note that similar errors (refer to Appendix D) were also made by humans during the process of annotation.

## 7.4 Comparison with SummaC-Conv

To analyze the performance of SummaC-Conv, we reproduced the model and ran it on the test set of FactCC (by following their paper Laban et al. (2022b)). We perform an error analysis to understand where the SummaC-Conv model makes errors. We find that out of 504 instances in the test set of FactCC dataset, SummaC-Conv model makes erroneous predictions in 75 instances. We categorize these errors into 5 categories: Exact claim, Paraphrased clain, Ambiguous claim, Coreference in claim and Incorrect true label. We describe these categories with examples for each error category in Figure 5. Table 3 shows the distribution of these categories. The last column of Table 3 shows in the number of error instances in each category in which our model performs correctly. It can be seen that our model shows superior performance in all categories. The majority of the error cases (∼71.6%) in SummaC-Conv belong to the category of "Ex-

| Error Category | Example | SummaC -Conv | Our Model | True Label |
|---|---|---|---|---|
| **Exact claim**: when claim is present in the context as is | [CLS] hillary clinton announced she was running for president . [SEP] ( cnn ) larry upright died just one day after hillary clinton announced she was running for president . [SEP] | H | NH | NH |
| **Paraphrased claim**: when a paraphrased version of claim is present in the context | [CLS] dozens of attackers took control of government buildings , officials say . [SEP] dozens of attackers took control of government buildings , including the city ' s central prison , central bank and radio station during the assault early thursday , according to officials . [SEP] | H | NH | NH |
| **Ambiguous claim**: when claim contains semantically ambiguous information | [CLS] mark the anniversary of a tragic event , the lingering memories of the 2013 marathon blasts will be seen and felt in all sorts of ways . [SEP] as the boston marathon runners begin and , hopefully , finish their exhausting run monday , the lingering memories of the 2013 marathon blasts will be seen and felt in all sorts of ways . [SEP] | H | NH | NH |
| **Coreference in claim**: when claim contains a reference to some entity in context | [CLS] peshmerga forces are national military force of kurdistan . [SEP] the peshmerga are the national military force of kurdistan . [SEP] | H | NH | NH |
| **Incorrect true label**: when the true label in the test data is itself incorrect | [CLS] hundreds of migrants on board may have capsized after being touched or swamped by a cargo ship that came to its aid . [SEP] catania , italy ( cnn ) the boat that sank in the mediterranean over the weekend with hundreds of migrants on board may have capsized after being touched or swamped by a cargo ship that came to its aid , a u . n . official said . [SEP] | NH | NH | H |

**Figure 5:** The five different types of scenarios in which SummaC-Conv gives incorrect predictions. An example for each category is shown along with the predictions made by SummaC-Conv and Our Model. Here, the darker colours depict higher importance scores while the light colours depict lower importance scores. The format is [CLS] <claim> [SEP] <context> [SEP]. The last column shows the true label, i.e. H: Hallucinated and NH: Non-hallucinated.

act claim" and "Paraphrased claim". Our model gives the correct prediction in 52 out of 53 of these cases, which is due to the global attention bias module that enables our model to focus on the relevant parts of the context and give correct output whereas SummaC-Conv lacks that ability. Similarly, in the cases of ambiguous claims or co-reference, our model performs better. The last error category, *Incorrect true label*, includes 8 cases where the true label is itself found to be wrong. Here, in 5 out of 7 cases, we find that our model gives the same prediction as the true label. Figure 5 shows examples for each error category, with the heat maps of our model for each example. The heat maps depict the local attention bias scores generated by our model where more scores to the necessary tokens, which helped in making a correct prediction.

## 8 Conclusion and Future Work

Hallucination detection is a pertinent and unresolved task in computational linguistics, often necessitating the incorporation of extralinguistic information into models. While traditional approaches benefit from leveraging diverse knowledge sources, we recognize the limitations of relying on outdated information and propose a rather orthogonal and cognitive-inspired approach that harnesses human eye-gaze patterns for hallucination detection. To support our approach, we created and analyzed an eye-tracking corpus, which revealed various attention biases exhibited by humans during hallucination searches. Building upon this insight, we designed a deep learning architecture that combines gaze data and hallucination classification data for training. Our experiments demonstrate promising results, indicating that a gaze-driven model exhibits superior performance and improved interpretability. Moving forward, we intend to explore deeper cognitive features, such as progressive and regressive saccades, to further enhance hallucination detection. Additionally, extending this work to multilingual and multimodal settings is a key objective on our agenda.

## Limitations

The existing cognitive framework falls short when it comes to addressing situations of internal inconsistency where multiple sentences within a given context contradict one another. Moreover, the current research overlooks intricate linguistic phenomena such as sarcasm and thwarting, focusing exclusively on hallucinations that emerge from perturba-

tions. Recognizing the diverse origins of hallucinations is crucial, and effectively addressing this challenge may require datasets encompassing a wider range of real-world scenarios and pragmatic instances.

## Ethics Statement

Conducting eye-tracking experiments for hallucination detection tasks requires a commitment to ethical principles, respect for participants' rights and well-being, and the responsible use and reporting of the collected data. To this end, we took several steps. Prior to conducting such experiments, we obtained informed consent from participants, clearly explained the purpose, procedures, and potential risks involved. Privacy and confidentiality was ensured by safeguarding participants' personal information and anonymizing their data. To minimize any discomfort or potential harm to participants during the eye-tracking sessions, adequate breaks and measures to prevent eye strain were provided. Participants were informed about their right to withdraw from the study at any point without penalty or consequence. After collection, participants' eye movement data have been treated with utmost respect and handled in accordance with applicable data protection regulations. It is also worth noting that while selecting examples from the FactCC dataset for the eye-tracking experiments, we strived for diversity and inclusivity to avoid perpetuating or reinforcing any social, cultural, or gender biases.

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

# A Global Attention Bias

## A.1 Models Used for Ensembling

As mentioned in Section 4.1.1, we employ an ensemble approach using the following sentence-transformer models: *all-roberta-large-v1* (Liu et al., 2019), *all-mpnet-base-v1* (Song et al., 2020), *gtr-t5-large* (Ni et al., 2021), *all-mpnet-base-v2* (Song et al., 2020), and *LaBSE* (Feng et al., 2020). Table 4 describes the model sizes and performances[7] on generating sentence embeddings.

| Model | Performance | Model Size |
|---|---|---|
| all-roberta-large-v1 | 70.23 | 1360 MB |
| all-mpnet-base-v1 | 69.98 | 420 MB |
| gtr-t5-large | 69.90 | 640 MB |
| all-mpnet-base-v2 | 69.57 | 420 MB |

**Table 4:** Model sizes and performance on generating sentence embeddings for different sentence-transformer models. The performance is the average performance (%) on encoding sentences over 14 diverse tasks from different domains.

The LaBSE model is a multilingual embedding model but shows good Semantic Textual Similarity

---

[7]Source: https://www.sbert.net/docs/pretrained_models.html

| Approach | Accuracy |
|---|---|
| **Voting** | **0.9608** |
| Only LaBSE | 0.9589 |
| Linear Regression | 0.9512 |
| Uniform Distribution | 0.9501 |

**Table 5:** Accuracy of various ensembling mechanisms used to determine the most accurate way to simulate human global attention bias.

(STS) benchmark (Cer et al., 2017) performance (72.8) as measured by Pearson's *r*.

## A.2 Implementation of Global Attention Bias

Section 4.1.1 discussed how global attention bias is modelled. Figure 6 shows a visualization of this modelling indicating how the global attention bias model has been incorporated into our framework. Table 5 shows the accuracy of various ensembling mechanisms which we explored to determine the most accurate way to simulate human global attention bias.

## B    Eye tracking Experiment for IITB-Hallucination Gaze corpus

### B.1    Experimental and Environmental Settings

We use the SR Research Experiment Builder software[8] for the eye-tracking data collection experiment. It should be noted that while sampling instances from the FactCC dataset, we ensured minimal variance in the length of selected instances. During data acquisition, careful measures were taken to minimize sound and light interference in the room. Participants were positioned 70 cm away from a 24-inch BenQ XL2420Z Widescreen LED Backlit TN Monitor, which had display dimensions of 569x337.8mm and a resolution of 1920x1080 pixels, resulting in a display area of 531.36x298.89mm. The monitor had a vertical refresh rate of 144 Hz. Sentences were presented in black font on a light grey background using a 20-point Arial font, equating to a letter height of 0.8mm. To ensure stability, a chin rest was provided, and participants were instructed to maintain their head position throughout the experiment to avoid motor artefacts. Refreshments and breaks were provided to participants to promote relaxation between iterations.

---

[8]http://www.sr-research.com/

Throughout the annotation process, eye position, and pupil size were meticulously tracked using an infrared video-based eye tracker (Eyelink 1000 plus Head Supported Version 5.03, SR Research) at a sampling rate of 2000 Hz for both eyes. The eye tracker had an instrumental spatial resolution of 0.01° and a microsaccade resolution of 0.05°. Before each iteration, the eye tracker was calibrated using a 9-point grid. Participants were instructed to sequentially fixate their gaze on the dot presented at each of the nine locations in random order. The calibration process was repeated until the discrepancy between two measurements at any point was below 0.5°, or the average error across all points was less than 1°.

### B.2    Annotation Guidelines

The instructions outlined a task that involves reading a claim and determining its faithfulness or consistency with respect to a given context. Participants were instructed to press the "+" key if the claim was faithful and the "-" key if unfaithful. The task emphasized the importance of careful reading and attention to detail. Before starting the annotation session, participants were asked to count the times the letter "F" occurred in a given text as a warm-up exercise. This exercise was intended to ensure that participants read the upcoming instances carefully. During the annotation session, participants were presented with examples of claims and corresponding contexts. They were expected to evaluate the faithfulness of each claim based on the highlighted text in the given context. Participants were reminded that their judgments should be based on the provided context and not require additional information. The examples provided demonstrated the expected annotation process. In Example 1, the claim accurately reflected the information given in the context, leading to an accurate annotation. In Example 2, the claim contradicted the information in the context, resulting in an unfaithful annotation. Participants were encouraged to maintain efficiency regarding time and accuracy while annotating. The task was to be approached as a search and matching exercise, focusing on the highlighted text within the context. Once participants familiarized themselves with the instructions and examples, they could click to start the annotation session.

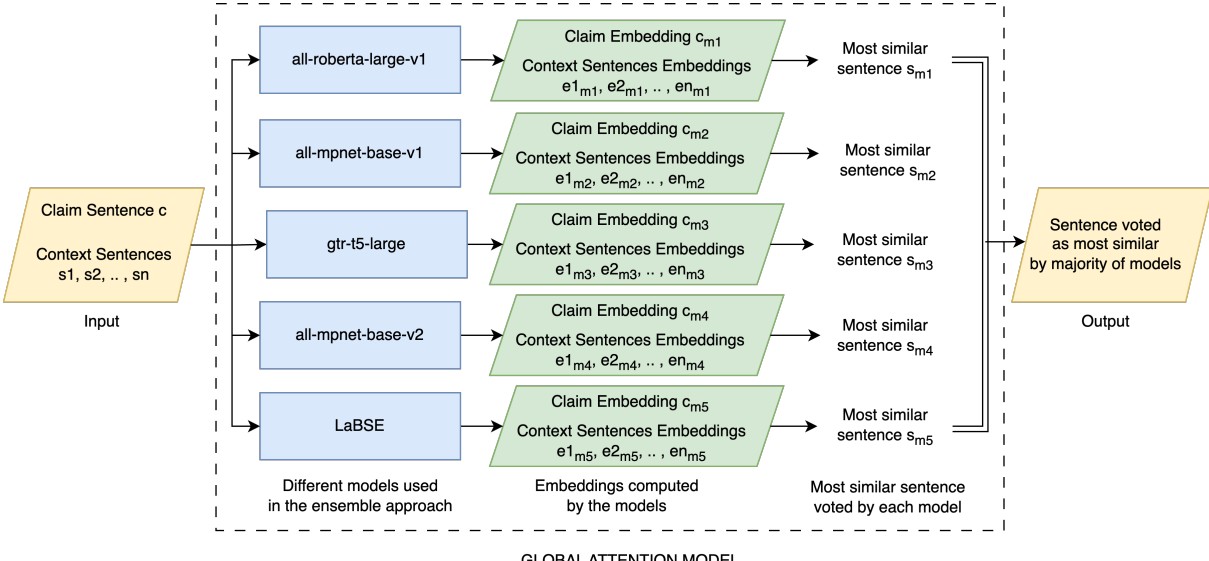

**Figure 6:** Modelling global attention bias using an ensemble approach. Each model m1,m2,...,m5 computes sentence embeddings for claim and context sentences. The sentence which is most similar to the claim is found by using cosine similarity between the claim embedding and sentence embedding for each model. The context sentence voted as most similar by the majority of models is taken as the output of the global attention model.

## C    Baselines Definition

In Section 6 we show a comparison study of the results of our model with other baseline models. The details of those baseline models are given in this section. We have considered the following models as a baseline for comparison defined in (Luo et al., 2023) for the given task.

- BERT + MNLI (Kryscinski et al., 2020): BERT trained on MNLI dataset (Williams et al., 2018)

- BERT + FEVER (Kryscinski et al., 2020): BERT trained on FEVER dataset (Thorne et al., 2018)

- MNLI-doc (Liu et al., 2019): finetunes Roberta model on MNLI dataset and incorporates entailment-based approach for hallucination detection

- NER Overlap (Laban et al., 2021): incorporates named entity matching from claim to context

- FACTCC-CLS (Kryscinski et al., 2020): Roberta model finetuned on FACTCC dataset

- DAE (Goyal and Durrett, 2020): incorporates NLI based approaches in dependency arc for detecting hallucination

- FEQA (Durmus et al., 2020): incorporates question answering based approach to detect the inconsistency from the extracted answers from document summary pair.

- QuestEval (Scialom et al., 2021): incorporates QA based metrics in the FEQA approach.

- ChatGPT-ZS(Luo et al., 2023): Chat-GPT in zero shot setting

- ChatGPT-COT (Luo et al., 2023): Chat-GPT in chain of thought setting (Wei et al., 2022)

- SummaCZS (Laban et al., 2022b): incorporates the NLI-based approaches for generating entailment scores at sentence level granularity followed by statistical aggregations like mean, median etc.

- SummaC-Conv (Laban et al., 2022b): incorporates convolution layer for aggregation in the previous approach

## D    Human Error evaluations

In this section, we provide a further example in Figure 7 to describe the case of semantic overlap.

The actual label for the specific claim-context pair provided in Figure 7 is "Hallucinated," while the prediction by the human annotator is "Not Hallucinated." This mismatch can be attributed to the

| Task | Dataset | Tokens (words) |
|------|---------|----------------|
| Natural Reading | Zuco 1.0 (Hollenstein et al., 2018) | 21,629 |
| Natural Reading | Dundee (Kennedy et al., 2013) | 56,212 |
| Natural Reading | PSC (Kliegl et al., 2004) | 1,138 |
| Natural Reading | GECO (Cop et al., 2017) | 114,080 |
| Natural Reading | Provo (Luke and Christianson, 2018) | 2,689 |
| Natural Reading | CopCo (Hollenstein et al., 2022) | 34,897 |
| Natural Reading + Relation Extraction | Zuco 2.0 (Hollenstein et al., 2020) | 15,138 |
| Sarcasm Detection | Sarcasm corpus (Mishra et al., 2016) | ~25,000 |
| Cognate Detection | Cognate Corpus (Kanojia et al., 2021) | 3,195 |
| **Hallucination** | **IITB-Hallucination Gaze corpus*** | **52,000** |

**Table 6:** Summary of Gaze Datasets and Word Counts for Various NLP Tasks

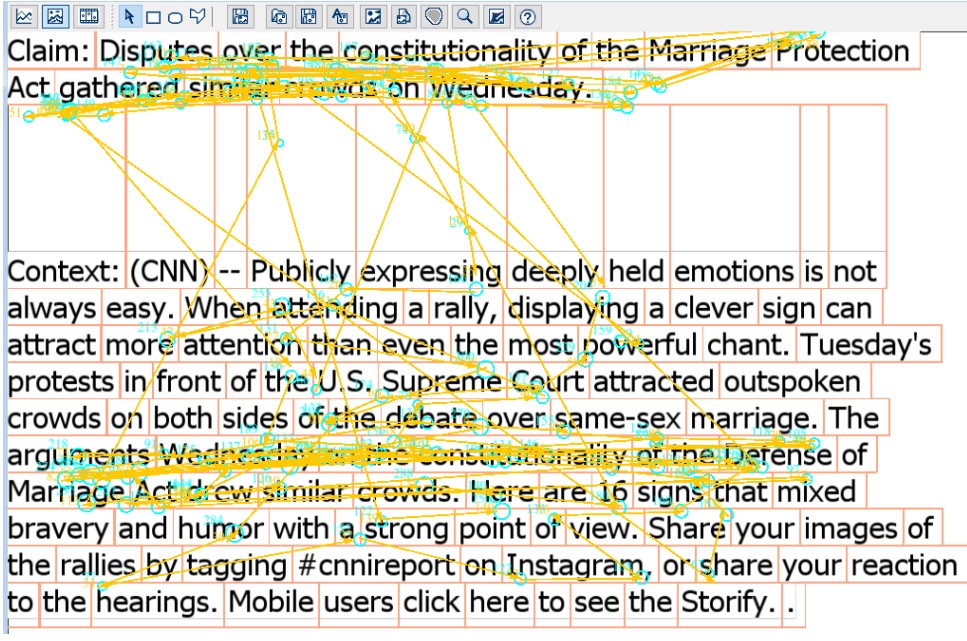

**Figure 7:** Example showing a higher semantic similarity between the claim and the context. For this example, the true label is hallucinated and the predicted label by the annotator is non-hallucinated.

considerable semantic similarity between the "Marriage Protection Act" of the claim and the "Defence of Marriage Act" of the context, leading to the annotator's prediction of "Not Hallucinated."

Claim: Disputes over the constitutionality of the Marriage Protection Act gathered similar crowds on Wednesday.

Context: (CNN) – Publicly expressing deeply held emotions is not always easy. When attending a rally, displaying a clever sign can attract more attention than even the most powerful chant. Tuesday's protests in front of the U.S. Supreme Court attracted outspoken crowds on both sides of the debate over same-sex marriage. The arguments Wednesday on the constitutionality of the Defense of Marriage Act drew similar crowds. Here are 16 signs that mixed bravery and humor with a strong point of view. Share your images of the rallies by tagging cnnireport on Instagram, or share your reaction to the hearings. Mobile users click here to see the Storify. .

# E   Comparison with Other Datasets

As mentioned in the Dataset Details section 5.1, we provide a word-level comparison with other publicly available gaze datasets.