# OpenReview forum: "Eyes Show the Way: Modelling Gaze Behaviour for Hallucination Detection"
_EMNLP/2023/Conference — EMNLP 2023 Findings_

### Official Review · Reviewer_WXZX · 2023-08-03

**Soundness:** 3

**Excitement:**

3: Ambivalent: It has merits (e.g., it reports state-of-the-art results, the idea is nice), but there are key weaknesses (e.g., it describes incremental work), and it can significantly benefit from another round of revision. However, I won't object to accepting it if my co-reviewers champion it.

**Paper Topic And Main Contributions:**

This paper proposes to leverage gaze information to attend more appropriately for
detecting hallucination on the output of large language models. Specifically, first the
most relevant sentence is selected from global attention, also obtained from gaze
information, and afterwards local attention gives a fixation score for each word to
be fed into a hallucination detector neural network.
Experimental results indicate that the accuracy of the final model is slightly worse than
the state-of-the-art, but basically equally well. Simple example is given for anecdotal
explanation.

Basically the idea is interesting and works well, it is a bit pity that the property of
the proposed approach is not well investigated. Since the performance itself does not
seem to	be a final goal	(since sota model performs better), *where* and *why* the gaze
information works is the most important for this research. For that purpose, a simple
anecdotal example in Section 7 is clearly not enough, and some systematic comparison
needs to be executed. If the gaze information is shown to be somewhat orthogonal to the
information used in non-gaze approaches, the impact of this paper will be larger
upon publication.


**Reasons To Accept:**

Showing that the gaze information is beneficial for detecting hallucination.

**Reasons To Reject:**

The paper does not show *how* the gaze information is different from ordinary attention for detecting hallucinations.

**Reproducibility:**

4: Could mostly reproduce the results, but there may be some variation because of sample variance or minor variations in their interpretation of the protocol or method.

**Reviewer Confidence:**

3: Pretty sure, but there's a chance I missed something. Although I have a good feel for this area in general, I did not carefully check the paper's details, e.g., the math, experimental design, or novelty.

---

> ### Author Rebuttal · Authors · 2023-08-29
>
> Dear esteemed reviewer, We thank you for your reviews.
>
> We've analyzed our experiments in Section 7, revealing that our system (BERT+GAB+LAB+gaze) significantly enhances interpretability. Notably, comparing Figure 4a (BERT+GAB+LAB finetuned on gaze) with Figure 4c (BERT+GAB based on raw attention) highlights the interpretability improvement from incorporating gaze. This underlines the distinction between gaze information and standard attention in hallucination detection.
>
> We have done similar analysis on other instances as well which show a similar trend. We plan to include in the camera-ready version if accepted.
>
> Here are some additional insights on interpretability analysis:
>
> -   In non-hallucinated cases, we notice high attention on identical words in both the claim and context, which aligns with the idea that shared semantics between them promote consistency.
>
> -   In hallucinated instances, we observe that certain parts of the context that were perturbed to form a hallucinated claim are given more importance scores.
>
>
> Here are some points which highlight the utility of the proposed cognitive framework in contrast to traditional methods:
>
> Our proposed attention bias framework: global (sentence level importance) and local (token level importance) is inspired by human behaviour. We emphasize that only local attention bias can be contrasted with ordinary attention, as the latter lacks the concept of sentence-level significance.
>
> The utility of global and local attention biases is explained below:
>
> -   **Global attention bias** focuses on relevant sentences, eliminating noise from inputs. In section 3.3 (line 302), we highlight that **only 20-30% of tokens per instance** are required on average for inference due to global attention bias. This significantly reduces the computational requirement and enables the use of LLMs trained for a low number of input tokens. This can also help other NLP tasks with high input sizes.
>
>
> -   **Local attention bias** (boosts importance scores of the words): Figure 4 shows that LAB improves interpretability compared to vanilla architecture. Fine-tuning LAB with gaze induces inductive bias gathered from human behaviour resulting in better and more natural token level interpretability. Consequently, analyzing local attention also yields richer psycholinguistic insights.

---

### Official Review · Reviewer_Qrwq · 2023-08-04

**Soundness:** 3

**Excitement:**

4: Strong: This paper deepens the understanding of some phenomenon or lowers the barriers to an existing research direction.

**Paper Topic And Main Contributions:**

This paper focuses on hallucination detection and introduces a novel cognitive approach that utilizes gaze signals from humans.

The authors begin by collecting a valuable eye-tracking corpus comprising 500 instances, which serves as the foundation for their proposed hallucination detection method.

They demonstrate the consistency of human behavior and attention bias phenomenon in psychology, providing insights that support the effectiveness of their approach.




**Questions For The Authors:**


1. Have you considered exploring the integration of gaze information with trending open-source LLMs for hallucination detection in your future research? How do you think gaze signals could potentially augment the capabilities of LLMs in detecting hallucinations?

2. In Section 5.1.3, could you clarify the differences between your proposed dataset and the other published datasets? What are the specific characteristics that set your dataset apart from the others?

3. Do you think it would be beneficial to adopt a multi-modal approach for hallucination detection, incorporating gaze information with other modalities like text or audio? How do you envision a multi-modal approach enhancing the performance of hallucination detection models?

**Reasons To Accept:**

1) This paper shares a valuiablel eye-tracking corpus, this may be very helpful for this task in the future.

2) The proposed method seems sound and effective to some extent.



**Reasons To Reject:**


The analysis section is relatively brief and lacks depth, requiring a more comprehensive and thorough analysis.

The corpus comprising only 500 instances might be considered small for certain tasks, raising concerns about the generalizability and statistical significance of the findings.

**Reproducibility:**

3: Could reproduce the results with some difficulty. The settings of parameters are underspecified or subjectively determined; the training/evaluation data are not widely available.

**Reviewer Confidence:**

3: Pretty sure, but there's a chance I missed something. Although I have a good feel for this area in general, I did not carefully check the paper's details, e.g., the math, experimental design, or novelty.

---

> ### Author Rebuttal · Authors · 2023-08-29
>
> Dear esteemed reviewer, thank you for your reviews.
>
> Here are a few points addressing the analysis:
>  -   We have performed an analysis of the experimental outcomes as mentioned in Section 7 which shows our system (BERT+GAB+LAB+gaze) has the capability of much better interpretability (see Figure 4). We have also addressed error scenarios in the Limitations section of the paper. Apart from this, we also show a behavioural analysis of humans while they perform the task of hallucination detection in section 3.3 providing significant psycholinguistic insights.
>
>  -   Here are some of the additional analyses that provide further insights:
> 	 - Interpretability:
>
> 		- In non-hallucinated instances, we notice that identical words appearing in both the claim and context are strongly focused on. This suggests that the claim and context share similar meanings. This aligns with the idea that similar semantic presence in claim and context leads to consistency.
> 		- In Hallucinated instances, we observe that perturbations of the context responsible for hallucinations are given more importance scores.
>
> 	 - Error Analysis:
> 		 - We observe an **overall error rate of 10.44%**.
> 		 - We observe that only **0.4%** of instances were misclassified as hallucinated due to semantic inconsistency in the instances.
> 		 - We observe that **10.04%** of instances were misclassified as non-hallucinated due to high semantic overlap between claim and context as mentioned in the Limitation section.
>
>
> Here are a few points that justify the size of the data:
>
> -   We humbly state that the purpose of gaze data is to do a behavioural analysis of humans during the task of hallucination detection. This makes 500 instances enough to gather and validate the insights from this study which were used to design the proposed cognitive framework and ground the global attention bias model.
>
> -   For finetuning the local attention bias model, we have added additional sentence-level fixation scores taken from Provo and Geco Corpus making a total of 5673 instances making it sufficient for the stated purpose.
>
>
> -   Here is a word-level comparison with other publicly available gaze data sets.
> | Task                        | Dataset               | Tokens       |
> |-----------------------------|-----------------------|--------------|
> | Natural Reading | Zuco 1.0              | 21,629 words |
> | Natural Reading + Relation Extraction         | Zuco 2.0              | 15,138 words |
> | Sarcasm Detection           | Mishra et al. (2016)  | ~25000 words |
> | Natural Reading              | Dundee                | 56,212 words |
> | Cognate Detection           | Kanojia et al. (2021) | 3,195 words   |
> | **Hallucination**               | **Ours***                 | **52,000 words** |
>
>
>
> **Response to “Questions For the Authors”**
>
> 1.  Yes, we have considered the integration of gaze information in the trending LLMs. This is one of the items on our next agenda.
> The gaze information is highly beneficial for less powerful LLMs that lack proper semantic/pragmatic understanding. Essentially the purpose of gaze in LLMs is to induce inductive bias towards certain prior which can act as a surrogate to world knowledge.
> LLMs with large parameter space can benefit from gaze to enhance explainability or instil human behaviour in the model when instances requiring human judgment are considered.
>
>
> 2.  In the context of the proposed architecture, the FactCC dataset has only textual instances for hallucination (without gaze) and the PROVO+GECO dataset has textual instances for the natural reading task along with gaze. Our contributed dataset **has gaze features along with textual instances for hallucination**.
>
>     Here are some more details pointing out the differences in our dataset from other publicly available eye-tracking datasets:
>     -   The prior published eye-tracking datasets mostly consider natural reading tasks and seldom delve into task-specific cases like sarcasm detection, sentiments etc. To the best of our knowledge, **no prior dataset is available for hallucination detection**.
>     -   Moreover, this dataset would find its utilities in **information retrieval tasks** as the given task involves searching for relevant context and then matching the same with the given claim. A similar kind of behaviour can be expected in the task of information retrieval.
>
>
> 3.  Yes, we think the use of a multimodal approach can be beneficial for hallucination detection. Audio and video can provide important information like **tonal behaviour (pauses may characterize inconsistency)** and **facial expressions (surprise)** which can be important indicators of inconsistency. We believe that these signals can significantly benefit very niche sets of instances that can not be disambiguated from textual information.

---

### Official Review · Reviewer_EMNe · 2023-08-05

**Soundness:** 3

**Excitement:**

3: Ambivalent: It has merits (e.g., it reports state-of-the-art results, the idea is nice), but there are key weaknesses (e.g., it describes incremental work), and it can significantly benefit from another round of revision. However, I won't object to accepting it if my co-reviewers champion it.

**Paper Topic And Main Contributions:**

This is a paper about hallucination detection. In this paper, In order to address the sustainability issue arising from traditional methods that require the latest knowledge, this paper proposes a novel approach to hallucination detection by leveraging the gaze pattern in the form of human cognitive and behavioral information. The main contributions of this paper are as follows:
1.	This paper creates and shares the first eye-tracking corpus for hallucination detection, which includes 500 instances of context and claim pairs annotated by five annotators.
2.	This paper's behavioral analysis of the annotated data reveals a recurrent pattern where annotators tend to browse through some irrelevant contexts while selectively focusing on information crucial for establishing or refuting hallucinations. By analyzing the gaze patterns of human annotators during the hallucination detection task, a novel concept of attention bias is introduced.
3.	This paper introduces and evaluates a BERT-based cognitive-inspired deep learning framework for hallucination detection driven by attention biases in various forms observed during human reading. Experimental evaluations on the FactCC dataset demonstrate the effectiveness of our approach, surpassing baseline models while achieving improved interpretability.

**Questions For The Authors:**

A.	The figure 2 in the paper is too simplistic, and the axes are not consistent, making it visually unappealing. The author may consider revising the figure for better presentation. Similarly, in figure 3, the arrows and symbols representing clicks have some issues. The author could optimize these two figures for better clarity and visual appeal.
B.	In the results section, there is indeed a lack of necessary analysis of the experimental outcomes. It is hoped that the author can provide additional explanations and clarifications.
C.	In the results section, while some ablation results are included, they are not sufficiently detailed. The author could further conduct more comprehensive ablation experiments, such as excluding the use of global attention and local attention and observing the results with only “BERT+GAZE”.

**Reasons To Accept:**

1.	The innovation in this paper is remarkable. Through analysis, it is evident that humans selectively focus on relevant parts of the text based on distributional similarity, akin to the phenomenon of attention bias in psychology. This paper identifies two attention strategies employed by humans: global attention (focusing on the most informative sentences) and local attention (focusing on important words within sentences). Leveraging these insights, the paper proposes a novel cognitive framework for hallucination detection that integrates these attention biases derived from human gaze data.
2.	The experiments in this paper are comprehensive and effectively validate the effectiveness of the human eye-gaze patterns for hallucination detection.
3.	This paper collects the first eye-tracking data from five annotators, annotating 500 instances of claim-context pairs carefully extracted from the FactCC dataset. During the annotation process, the paper captures consistent patterns of the annotators' gaze on the claim and context texts, along with their respective labels. Notably, the inter-annotator agreement (IAA) Kappa score reached 0.60, indicating substantial agreement among the annotators.

**Reasons To Reject:**

1.	The experimental results in this paper were obtained using a dataset constructed by the authors, and there is no validation on other datasets to verify the reliability of the proposed method.
2.	In the results section, the author merely lists their experimental results and the compared baselines without conducting further analysis on the experimental outcomes. Additionally, the explanations for these baselines in the appendix are overly simplistic.
3.	In the experimental results of Table 3, the baseline "SummaC-Conv" achieves higher performance than the results presented in the paper. However, the author does not provide any explanation or interpretation.

**Reproducibility:**

3: Could reproduce the results with some difficulty. The settings of parameters are underspecified or subjectively determined; the training/evaluation data are not widely available.

**Reviewer Confidence:**

2: Willing to defend my evaluation, but it is fairly likely that I missed some details, didn't understand some central points, or can't be sure about the novelty of the work.

---

> ### Author Rebuttal · Authors · 2023-08-29
>
> Dear esteemed reviewer, thank you for the reviews.
>
> We would like to clarify a few points -
>  -   The experimental results were obtained on **FactCC** (section 5.1.1: line 441), a dataset **publicly released by Kryscinski et. al. (2020)**. Our contribution was to add gaze signals to this dataset which was used in our framework (only to train local and global attention). We specifically relied on FactCC (section 5.1.1: line no.448) due to the presence of a broad range of claim-context relationships and well-defined perturbation crafted by introducing variations in the context which facilitated the eye-tracking experiments.
>  -   We have performed an analysis of the experimental outcomes as mentioned in Section 7 which shows that the proposed framework has the capability of much better interpretability (see Figure 4). We have also addressed error scenarios in the Limitation sections of the paper. Apart from this, we also show a behavioural analysis of humans while they perform the task of hallucination detection in section 3.3 which provides significant psycholinguistic insights.
>  -   Here are some of the additional analyses that provide further insights:
>
> 	 - Interpretability:
> 		 - In non-hallucinated instances, we notice that identical words appearing in both the claim and context are strongly focused on. This suggests that the claim and context share similar meanings aligning with the idea that similar semantic presence in claim and context leads to consistency.
> 		 - In Hallucinated instances, we observe that perturbations of the context responsible for hallucinations are given more importance scores.
> 	 - Error Analysis:
> 		 - We observe an **overall error rate of 10.44%.**
> 		 - We observe that only **0.4%** of instances were misclassified as hallucinated due to semantic inconsistency in the instances.
> 		 - We observe that **10.04%** of instances were misclassified as non-hallucinated due to high semantic overlap between claim and context as mentioned in the Limitation section.
>
> 	 But we humbly ask for further clarification about what the reviewer exactly means by “further analysis of experimental outcomes” and what additional details are requested in the explanation of baselines.
>
> -   Regarding the performance of the SummC-Conv model: the FactCC dataset is imbalanced and noisy, the SummaC-Conv binning strategy provides the architecture with good noise reduction capability resulting in a slightly better balanced accuracy score.
>
> Response to “Questions For the Authors”
>
> 1.  Thank you for pointing this out, we will improve the figures and make them more consistent. It would be great if you could provide more input on how to make the figure more visually appealing.
>
> 2.  As mentioned before, we provide analysis in Section 7 and we talk about the error scenarios in the Limitations section and in the appendix. We also conducted some further analysis and reported the same above in this response
>
> 3.  In our architecture, it is **not feasible to integrate the “GAZE” signal** directly with the BERT architecture since it requires a local attention bias / global attention bias module, contrary to the other architectures mentioned in section 2.2 (line no.196 to 210).  The gaze signal is required to ground the Global attention bias (for sentence level importance: line no. 349 to 356) and finetune the Local attention bias model (for token level importance: line no. 395 to 400). Hence “GAZE” signal has no direct interaction with the BERT, therefore we have not reported GAZE+BERT ablation results.
>
> ***
> Thanks a lot for all the valuable feedback. Should our response meet your expectations, we kindly request your consideration in the possibility of favorably adjusting the scores.

---

### Meta-Review · Area_Chair_L1vd · 2023-09-16

**Recommendation:** 1

**Metareview:**

The paper "Hallucination Detection Using Attention Biases Derived from Human Gaze Data" proposes a novel approach to hallucination detection by leveraging gaze patterns in the form of human cognitive and behavioral information. The main contributions of the paper include the creation and sharing of the first eye-tracking corpus for hallucination detection.

However, there are also several weaknesses as the reviewers raised. Firstly, the experimental results were obtained using a dataset constructed by the authors, and there is no validation on other datasets to verify the reliability of the proposed method. Secondly, the explanations for the baselines in the appendix are overly simplistic, and the authors merely list their experimental results and the compared baselines without conducting further analysis on the experimental outcomes. Thirdly, the paper does not provide a comprehensive analysis of the differences between the proposed dataset and the other published datasets, and it is unclear what specific characteristics set the dataset apart from the others.

A final note is that there are quite a few hallucination related studies released before the submission deadline of EMNLP 2023. But unfortunately, the authors seem to be unaware of these work,  and thus the novelty and reliability of this work can be questionable.

---

### Decision · Program_Chairs · 2023-10-07

**Decision:**

Accept-Findings

**Comment:**

The paper "Hallucination Detection Using Attention Biases Derived from Human Gaze Data" proposes a novel approach to hallucination detection by leveraging gaze patterns in the form of human cognitive and behavioral information. The main contributions of the paper include the creation and sharing of the first eye-tracking corpus for hallucination detection.

However, there are also several weaknesses as the reviewers raised. Firstly, the experimental results were obtained using a dataset constructed by the authors, and there is no validation on other datasets to verify the reliability of the proposed method. Secondly, the explanations for the baselines in the appendix are overly simplistic, and the authors merely list their experimental results and the compared baselines without conducting further analysis on the experimental outcomes. Thirdly, the paper does not provide a comprehensive analysis of the differences between the proposed dataset and the other published datasets, and it is unclear what specific characteristics set the dataset apart from the others.

A final note is that there are quite a few hallucination related studies released before the submission deadline of EMNLP 2023. But unfortunately, the authors seem to be unaware of these work,  and thus the novelty and reliability of this work can be questionable.